# Management of Central Precocious Puberty in Children with Hypothalamic Hamartoma

**DOI:** 10.3390/children8080711

**Published:** 2021-08-18

**Authors:** Junghwan Suh, Youngha Choi, Jun Suk Oh, Kyungchul Song, Han Saem Choi, Ahreum Kwon, Hyun Wook Chae, Ho-Seong Kim

**Affiliations:** Department of Pediatrics, Severance Children’s Hospital, Endocrine Research Institute, Yonsei University College of Medicine, Seoul 03722, Korea; suh30507@yuhs.ac (J.S.); younghachoi1986@gmail.com (Y.C.); joojang87@naver.com (J.S.O.); endosong@yuhs.ac (K.S.); hansaem6890@yuhs.ac (H.S.C.); armea@yuhs.ac (A.K.); hopechae@yuhs.ac (H.W.C.)

**Keywords:** hypothalamic hamartoma, central precocious puberty, gelastic seizure

## Abstract

Hypothalamic hamartoma (HH) is a rare, congenital, and benign lesion of the tuber cinereum, typically presenting with central precocious puberty (CPP), gelastic seizure, and developmental delay. This study aimed to investigate CPP in HH patients and compare clinical features between before and after gonadotropin-releasing hormone (GnRH) agonist treatment. A total of 30 HH patients under 18 years of age who visited Severance Children’s Hospital between January 2005 and May 2020 were retrospectively reviewed. Fourteen patients were male (46.7%) and sixteen (53.3%) were female, with a mean age at diagnosis was4.2 ± 2.9 years. During follow-up, 24 patients (80.0%) were diagnosed with CPP, 15 patients (50.0%) had gelastic seizure, and 13 patients (43.3%) had developmental delay. The gelastic seizure was significantly associated with sessile type HH rather than pedunculated type HH (85.7% vs. 18.8%, *p* = 0.001). After GnRH agonist treatment, discrepancies between bone age and chronological age decreased (3.3 ± 1.3 years to 2.0 ± 1.7 years, *p* = 0.002). Additionally, height standard deviation score for bone age was increased, and predicted adult height increased significantly in females, while males showed an increasing trend. Clinical symptoms of HH were closely associated with the location of HH, and GnRH agonist treatment was safe and effective in the management of CPP caused by HH.

## 1. Introduction

Hypothalamic hamartoma (HH) is a rare, congenital, and benign lesion of the tuber cinereum. It is mainly composed of normal brain tissue, such as neurons, glial cells, and fiber bundles [1]. The prevalence of HH varies between 1 in every 50,000 and 1 in every 200,000 persons [2,3,4]. The majority of HH is located at the base of the hypothalamus and the floor of the third ventricle [5]. It is usually diagnosed by brain magnetic resonance imaging (MRI) scans and is seen as non-enhancing, isointense, or hyperintense lesions on T2-weighted images [6,7].

HH is a complex neuroendocrine disease that usually first presents with one of the three following main symptoms; central precocious puberty (CPP), gelastic seizure, or developmental delay [8,9]. HH is one of the most common organic causes of CPP [10]. CPP caused by HH tends to occur significantly earlier in life than idiopathic CPP [11]. The pathophysiology of hypothalamus–pituitary–gonadal axis activation is not completely understood, but CPP in HH patients is well managed by treatment with a gonadotropin-releasing hormone (GnRH) agonist [12,13]. Gelastic seizure, also known as laughing seizure, is a distinct feature of HH [14]. Other various seizure types including generalized or focal seizures are reported in HH patients [15]. As antiepileptic drugs are frequently unsuccessful in seizures associated with HH, surgical intervention is often considered in intractable epilepsy [4,16]. Additionally, behavioral and cognitive disorders are often accompanied in HH patients with seizure events, resulting in developmental delay and diverse psychological problems [2,17].

Previously published research has reported case studies with HH, but most of these studies focused mainly on seizure control, neuroimaging, and surgical management. Previous studies focusing on CPP in HH are sparse, while studies concerning longitudinal follow-up data of CPP patients are especially rare. In this study, clinical characteristics, laboratory results, and image findings of HH are explored. Further, we investigated CPP in HH patients and compared clinical features between before and after GnRH agonist treatment.

## 2. Materials and Methods

### 2.1. Patients

A total of 30 patients under 18 years of age with HH who visited Severance Children’s Hospital between January 2005 and May 2020 were retrospectively reviewed. The diagnosis of HH was based on MRI scans. This study was approved by the Institutional Review Board of Severance Hospital, Yonsei University College of Medicine in Seoul, Korea (no. 4-2020-0241). The requirement to obtain informed consent was waived.

### 2.2. Study Design

Patient data, including sex, age at symptom onset, age at diagnosis, symptom at first visit, additional symptoms during follow-up, and type of received treatment, were reviewed. Additionally, brain MRI findings such as size and type of HH were gathered. CPP was diagnosed when the peak LH level was 5 ≥ IU/L on GnRH stimulation test before 8 years of age in girls and before 9 years of age in boys. In CPP patients, further clinical and biochemical data were collected, including age at diagnosis of CPP, height, weight, body mass index (BMI), serum luteinizing hormone (LH), follicle-stimulating hormone (FSH), prolactin, estradiol, testosterone, peak LH level on GnRH stimulation test, pubertal development stage, bone age, and duration of GnRH agonist treatment.

Brain MRI scans were acquired using a Philips 3.0T scanner (Philips Achieva 3.0T; Philips Medical Systems, Best, The Netherlands). Type of HH was determined based on brain location on MRI, and cases were classified into either pedunculated (parahypothalamic) type or sessile (intrahypothalamic) type [6,8,9]. Height standard deviation score (SDS) for chronological age, height SDS for bone age, weight SDS, and BMI SDS were calculated using the growth standard of Korean children and adolescents [18]. Serum LH, FSH, and prolactin levels were measured using sequential 2-step immunoenzymatic assays (Access Reagent Pack, Beckman Coulter Inc., Brea, CA, USA), with an intra-assay coefficient of variation (CV) of 3.5–5.4%, inter-assay CV of 4.3–6.4%, and a lower limit of detection of 0.2 IU/L for both gonadotropins. Serum E2 levels were measured using radioimmuno assays (Coat-A-Count Estradiol, Siemens, Erlangen, Germany), and serum testosterone levels were measured by chemiluminescent microparticle immunoassays (Architect i2000 analyzer, Abbott Diagnostics, Chicago, IL, USA). Pubertal development was determined using the Marshall and Tanner staging system [19,20]. Bone age was examined by performing radiography of the left hand and wrist and interpreted using the Greulich and Pyle (GP) method [21]. Predicted male and female adult height were estimated using the Bayley–Pinneau (BP) method [22]. All patients with CPP were treated with GnRH agonist every four weeks, either leuprolide acetate (3.75 mg) or triptorelin acetate (3.75 mg). GnRH agonist treatment was maintained until bone age reached 12 to 12.5 years in girls and 13 years in boys.

### 2.3. Statistical Analysis

Statistical analyses were performed using IBM SPSS ver. 25.0 (IBM Corp., Armonk, NY, USA). Anthropometric data, hormone levels, size of HH, tanner stage, bone age, and predicted adult height are presented as the mean ± SD. Fisher’s exact test was used to analyze the association between clinical symptoms and type of HH, and the paired t-test was applied to compare data before GnRH agonist treatment and after treatment. A *p*-value < 0.05 was considered statistically significant.

## 3. Results

### 3.1. Clinical Characteristics of Patients with HH

Among the total 30 HH patients, 14 patients were male (46.7%) and 16 patients were female (53.3%). Mean age at onset of first symptom was 2.9 ± 2.5 years, and age at diagnosis of HH was 4.2 ± 2.9 years. Twelve patients (40.0%) came to the hospital presenting precocious puberty as a first symptom, while nine patients (30.0%) had a gelastic seizure as the chief complaint for the first visit. Seven patients presented with other types of seizures, such as generalized tonic-clonic seizure (*n* = 3), simple partial seizure (*n* = 2), focal seizure (*n* = 1), and infantile spasm (*n* = 1). During the follow-up, 24 patients (80.0%) were diagnosed with CPP, 15 patients (50.0%) had gelastic seizure, and 13 patients (43.3%) had developmental delay. Twelve patients (40.0%) had surgical operation, and two patients (6.7%) underwent gamma-knife surgery. All 24 CPP patients received GnRH agonist treatment. Two patients were diagnosed with growth hormone deficiency by growth hormone stimulation test and received growth hormone treatment (Table 1).

### 3.2. Association between Clinical Manifestations and Type of HH

Type of HH was classified into pedunculated type or sessile type according to its location on brain MRI scans. Of the 30 patients, 16 patients (53.3%) had pedunculated-type HH while 14 patients (46.7%) had sessile type. Mean size of HH was 17.8 ± 9.2 mm (Table 1). Association between the type of HH and the classic triad of symptoms (precocious puberty, gelastic seizure, and developmental delay) is summarized in Table 2. Pedunculated type of HH had a higher prevalence of precocious puberty than sessile type (87.5% vs. 71.4%), but the difference was not significant (*p* = 0.378). Twelve of fourteen patients (85.7%) with sessile type HH had gelastic seizure, while only three of sixteen (18.8%) pedunculated-type patients presented gelastic seizures (*p* = 0.001). Developmental delay showed a preponderance of sessile type over pedunculated type, but the difference was not statistically significant (64.3% vs. 25.0%, *p* = 0.063).

### 3.3. Management of Central Precocious Puberty in HH Patients

Among the 24 CPP patients, 11 patients (45.8%) were male, and 13 patients (54.2%) were female. Mean age at diagnosis of CPP was 5.7 ± 3.1 years. Basal LH level was 2.48 ± 1.31 IU/L, estradiol was 25.74 ± 23.00 pg/mL, and testosterone was 227.26 ± 291.82 ng/dL. LH peak of GnRH stimulation test was 27.92 ± 18.94 IU/L. Breast tanner stage on diagnosis of CPP was 2.8 ± 0.6 and testis size was 12.1 ± 7.1 cc (Table 3).

Of the 24 CPP patients, 9 patients were referred to other hospitals, and the remaining 15 patients were continuously followed up in our clinic. Patients underwent GnRH agonist treatment for 3.1 ± 2.2 years on average (Table 3). Clinical findings of the 15 patients before and after GnRH agonist treatment are presented in Table 4. Height SDS for chronological age decreased after GnRH agonist treatment (1.38 ± 1.76 to 0.72 ± 1.38, *p* = 0.010), while height SDS for bone age significantly increased (−2.65 ± 1.76 to −1.43 ± 1.15, *p* = 0.003). Weight SDS and BMI SDS showed no significant change. Bone age was 3.3 ± 1.3 years advanced compared to chronological age at diagnosis, which significantly decreased to 2.0 ± 1.7 years after GnRH agonist treatment (*p* = 0.002). Predicted adult height significantly increased in females (149.0 ± 8.8 cm to 157.3 ± 9.7 cm, *p* = 0.006), while males showed an increasing trend (171.0 ± 12.4 cm to 175.2 ± 8.3 cm, *p* = 0.066).

## 4. Discussion

HH is a rare congenital disease only diagnosed by brain imaging, which makes it difficult to investigate its epidemiology such as prevalence, sex distribution, and clinical features. Some previous studies reported that HH is slightly more common in males, but the differences were not remarkable. Parvizi et al. reported that 59 of 100 (59.0%) HH patients were male [15], and Nguyen et al. found that 136 of 256 patients (53.1%) were male [4]. However, our study population showed nearly equal distribution among both sexes (14 male patients and 16 female patients). Additionally, HH is usually confirmed by brain MRI scans during the further evaluation of pubertal development or seizure events, which generally onsets in the early years in life. Corbet Burcher et al. reported that the mean age at symptom onset was 1.5 years based on a systematic review of 264 HH patients, and the mean age at diagnosis of HH was 2.4 years in their cohort with 46 HH children [17]. Our results showed a mean age of 2.9 years at symptom onset, and age at diagnosis of HH was 4.2 years, which is slightly older than previous studies [4,23,24].

HH has often been characterized by its distinct clinical symptoms. The majority of HH patients first visit hospitals complaining of early pubertal development or seizures. In our study, CPP was the most common first symptom, found in 12 patients, followed by gelastic seizure in 9 patients. However, considering that seven patients experienced other types of seizures at first visit, over half of the patients (16 of 30 patients, 53.3%) first presented with seizures. The prevalence of CPP in HH patients from our study is 80.0% (24 of 30 patients), which is relatively higher than in previous reports (28 of 67 patients, 41.7% [7], and 46 of 100 patients, 46.0% [15]). However, some other researches showed a higher prevalence of CPP in HH patients (174 of 277 patients, 62.8% [4], 115 of 214 patients, 53.7% [25]). Unlike other previous cross-sectional studies, we continuously followed up HH patients without CPP at first visit, and 12 patients had newly developed CPP during the observation. This longitudinal design of this study could have influenced differences in the prevalence of CPP in our HH patients.

HH is classified into either pedunculated type or sessile type based on its location in brain MRI scans. The association between the type of HH and clinical symptoms was investigated in many previous studies, showing a close relationship between pedunculated type HH and CPP. Further, a high correlation has been reported between the sessile type HH and gelastic seizure with accompanying developmental delay [4,8,24,25,26]. Our data showed similar results in terms of gelastic seizure, which was highly associated with sessile type HH. Additionally, CPP was more commonly diagnosed in the pedunculated type, but the results were not statistically significant. The developmental delay also showed a tendency toward sessile type HH.

Seizure in HH patients is often first treated with antiepileptic drugs, but the results are usually unsuccessful [15,27,28]. Surgical management is considered in patients with medically intractable seizure, which can make epileptic patients seizure-free and improve cognition and behavior problems. Various surgical techniques have been devised, such as microsurgical resection, endoscopic disconnection, stereotactic radiosurgery, radiofrequency ablation, and vagal nerve stimulation [29,30,31]. In our study, 19 of 30 patients had seizure history, while gelastic seizure was most commonly seen in 15 patients. Among the 19 seizure patients, 12 patients underwent surgical resection, and 2 patients underwent gamma-knife surgery. The mean timing of surgery was 5.7 years after initiation of pharmacological treatment. Among the 14 surgery patients, 8 patients were seizure-free after surgery and 6 patients had improvement of seizure activity. However, we did not observe any changes in seizure patterns in patients with GnRH agonist treatment. None of the 11 HH patients who had no seizure events received surgical intervention.

The mechanism of HH triggering the onset of puberty is not clearly understood, but there are a few proposed hypotheses. Previous studies suggested that HH controls GnRH secretion in several ways, such as producing ectopic GnRH within the HH itself or controlling nearby neurons synaptically connected to GnRH neurons [13,31,32]. Research by Chan et al. emphasized that the anatomic features of HH are more responsible for arousing pubertal development [33]. In any case, whatever the exact mechanism of the onset of puberty in HH patients may be, CPP caused by HH is well controlled by GnRH agonist treatment [12,23,34,35]. In our study, 15 CPP patients continuously received GnRH agonist treatment for an average of 3.1 years and had no suspected unexpected serious adverse reactions. All girls had their menarche at a mean duration of 17 months after discontinuation of GnRH agonist treatment. All 15 patients had follow-up brain MRI scans after starting GnRH agonist treatment, and there were no interval changes in HH size. We used three parameters to evaluate the efficacy of GnRH agonist treatment; the discrepancy between bone age and chronological age, height SDS for bone age, and predicted adult height by the BP method. After GnRH agonist treatment, bone age advancement was ameliorated by 1.3 years, and height SDS for bone age significantly increased. Additionally, predicted adult height by the BP method increased significantly in females, while males showed an increasing, but not statistically significant, trend.

There are some limitations to this study. Among the 24 CPP patients, 9 HH patients with CPP were referred to other hospitals, and the remaining 15 patients received GnRH agonist treatment continuously in our outpatient clinic. As GnRH agonist treatment is mainly based on monthly injection, patients prefer getting injections at hospitals near their residential area, resulting in a high proportion of follow-up loss. Additionally, we used the BP method to predict final adult height in HH patients with CPP. While the BP method is one of the most long-standing and widely used methods to predict adult height, it has some limitations. The BP method was established in the 1950s based on American children, which might be inadequate to apply directly to Korean children in the 21st century. Further, predicted adult height calculation by the BP method is not available in children with skeletal age (bone age) younger than six, and it is impossible to predict adult height in children of extremely high stature by the BP method. Lastly, 4 of 15 CPP patients are still under treatment of GnRH agonist, and 6 patients had surgical operation or gamma-knife surgery, which could have influenced our analysis. Further studies with a larger sample size and a longer follow-up period are required to elucidate more detailed results.

## 5. Conclusions

In summary, we investigated clinical characteristics, laboratory results, and brain MRI findings in 30 HH patients. Gelastic seizure was closely associated with sessile type HH, while CPP showed a tendency toward pedunculated type HH. In addition, based on 15 HH patients with CPP, we can reconfirm the safety and efficacy of GnRH agonist treatment in HH patients with CPP.

## Figures and Tables

**Table 1 children-08-00711-t001:** Characteristics of patients with hypothalamic hamartoma.

Characteristics	Value
Sex	
Male/Female	14 (46.7%)/16 (53.3%)
Age at symptom onset (year)	2.9 ± 2.5
Age at diagnosis (year)	4.2 ± 2.9
Symptom at first visit	
Precocious puberty	12 (40.0%)
Gelastic seizure	9 (30.0%)
Other seizures	7 (23.3%)
Headache	1 (3.3%)
Developmental delay	1 (3.3%)
Symptoms during follow-up	
Precocious puberty	24 (80.0%)
Gelastic seizure	15 (50.0%)
Developmental delay	13 (43.3%)
Treatment	
Surgical operation	12 (40.0%)
Gamma-knife surgery	2 (6.7%)
GnRH agonist treatment	24/24 (100%)
Growth hormone treatment	2 (6.7%)
Type of hypothalamic hamartoma	
Pedunculated/Sessile	16 (53.3%)/14 (46.7%)
Size of hypothalamic hamartoma (mm)	17.8 ± 9.2

Data are presented as mean ± SD or number of patients (%).

**Table 2 children-08-00711-t002:** Association between type of HH and symptom triad.

	Sessile Type	Pedunculated Type	*p* Value
Precocious puberty	10/14 (71.4%)	14/16 (87.5%)	0.378
Gelastic seizure	12/14 (85.7%)	3/16 (18.8%)	0.001
Developmental delay	9/14 (64.3%)	4/16 (25.0%)	0.063

**Table 3 children-08-00711-t003:** Characteristics of hypothalamic hamartoma patients with central precocious puberty (*n* = 24).

Characteristics	Value
Male/female	11 (45.8%)/13 (54.2%)
Age at diagnosis of CPP (year)	5.7 ± 3.1
Basal hormone levels at diagnosis	
LH (IU/L)	2.48 ± 1.31
FSH (IU/L)	4.18 ± 2.73
Estradiol (pg/mL) (*n* = 13)	25.74 ± 23.00
Testosterone (ng/dL) (*n* = 11)	227.26 ± 291.82
Prolactin (ng/mL)	12.95 ± 7.50
LH peak of GnRH stimulation test (IU/L)	27.92 ± 18.94
Breast tanner stage on diagnosis of CPP (*n* = 13)	2.8 ± 0.6
Testis size on diagnosis of CPP (cc) (*n* = 11)	12.1 ± 7.1
Duration of GnRH agonist treatment (year) (*n* = 15)	3.1 ± 2.2

CPP, central precocious puberty; LH, luteinizing hormone; FSH, follicle-stimulating hormone; GnRH, gonadotropin-releasing hormone. Data are presented as mean ± SD or number of patients (%).

**Table 4 children-08-00711-t004:** Comparison between before and after GnRH agonist treatment (*n* = 15).

	Before GnRH Treatment	After GnRH Treatment	*p* Value
Height (cm)	123.7 ± 19.4	141.0 ± 15.0	<0.001
Height SDS for chronological age	1.38 ± 1.76	0.72 ± 1.38	0.010
Height SDS for bone age	−2.65 ± 1.76	−1.43 ± 1.15	0.003
Weight (kg)	30.9 ± 12.4	44.8 ± 12.6	<0.001
Weight SDS	1.53 ± 1.17	1.41 ± 0.88	0.450
BMI	19.30 ± 2.88	22.29 ± 3.37	0.001
BMI SDS	1.23 ± 1.26	1.58 ± 0.94	0.214
Bone age (year)	9.9 ± 2.3	11.7 ± 1.9	<0.001
Chronological age (year)	6.6 ± 3.0	9.7 ± 2.7	<0.001
BA–CA (year)	3.3 ± 1.3	2.0 ± 1.7	0.002
Male predicted adult height (cm) (*n* = 6)	171.0 ± 12.4	175.2 ± 8.3	0.066
Female predicted adult height (cm) (*n* = 7)	149.0 ± 8.8	157.3 ± 9.7	0.006

GnRH, gonadotropin-releasing hormone; SDS, standard deviation score; BMI, body mass index; BA, bone age; CA, chronological age. Data are presented as mean ± SD.

## Data Availability

The data presented in this study are available on request from the corresponding author.

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
