# Peer review of "Management of Central Precocious Puberty in Children with Hypothalamic Hamartoma"

_children, 2021, doi:10.3390/children8080711_

Round 1
Reviewer 1 Report
In this paper, the authors investigated clinical characteristics, laboratory results, and brain MRI findings in 30 HH patients. They found that the gelastic seizure was closely associated with sessile type HH, while CPP showed tendency toward pedunculated type HH. In addition, based on 15 HH patients with CPP, they reconfirmed the safety and efficacy of GnRH agonist treatment in HH patients with CPP. Overall, this paper is written in professional English, with sufficient introduction, detailed methods and insightful discussion. I will encourage the authors to further elaborate the mechanisms of GnRH agonist in HH treatment and also to increase the sample size in their following studies.
Reviewer 2 Report
The submitted paper “Management of Central Precocious Puberty (CPP) in Children with Hypothalamic Hamartoma (HH)” presents a series of 30 patients who visited a tertiary Hospital in Korea from January 2005 to May 2020. Of those, 24 patients either presented or were diagnosed with CPP during their follow-up and 15 were subsequently followed in the same hospital.
As there are indeed very few case series with longitudinal data of patients with CPP due to HH, this is an interesting and clinically useful paper, that adds confirmatory data regarding the safety and efficacy of GnRH analogs in HH induced CPP, but also provides us with information as to the changes of height sds for chronological and bone age, the discrepancy between bone and chronological age, and the predicted adult height of the children.
There are, however some issues that need to be addressed by the authors.
- As the authors correctly mention, the percentage of patients with HH and CPP in their case series is rather high. They attribute this finding to the fact that they included not only CPP at diagnosis but also during the follow-up period. However, as they use the cut-off limit of 5 IU/ml for stimulated LH, I believe more info should be provided regarding the sequential 2-step immunoenzymatic assay (Access Reagent Pack, Beckman Coulter Inc., Brea, CA, USA) they used to perform the FSH/LH measurements.
- As the study period is rather long, I think that-if available- the authors could provide us with more data about the patients actually achieving final height, so as to be able to compare it with their predicted and target adult height. Moreover, if available, we could have more information about the re-activation of the axis and the actual age (or time from end of treatment) of first menstruation for girls. Is the follow-up period adequate, so as to be able to have this data?
- It is true that the duration of GnRH treatment seems a bit short, especially in comparison to other case series. The authors mention that only one patient is still under treatment. Could the authors comment on the timing of treatment withdrawal?
- In the case series, there was a significant reduction of height SDS for chronological age at the end of the treatment. Could this be attributed to possible growth acceleration prior to treatment and subsequent return to a more “normal for age” height after the treatment?
- The authors mention that there were 2 children that also received GH treatment. As there are only few reports on children with GH deficiency and HH, could the authors provide more information? Were they syndromic cases? Was GH deficiency secondary to surgery?
- In the Discussion, the authors repeat many of the results that have already been presents, either in the text or in the tables and only briefly make comparisons with previous published data. Given that there are so few case series available, a more in depth/ critical comparison of differences and similarities between the studies, as well as of their own findings could be interesting (especially since the authors have successfully referred to almost all published case series)
- Obviously, there were patients that had both seizures and CPP, so some of them must have undergone surgery. What was the effect of surgery on CPP? Are these patients included in the analysis of the outcome of the GnRH treatment?
- Since the authors have also included treatment of seizures in their discussion (although it is not their main endpoint), could they elaborate more as to the timing of surgery after the initiation of pharmacological treatment? Were they patients that were refractory to treatment or were they submitted to surgery immediately after diagnosis? Also, did they notice an improvement in seizures after the initiation of GnRH treatment (one case report)? What was the course of seizures after the surgery? (Obviously, since this is not the main point of the paper, these data are not necessary but could be added to the discussion, if available)
- Did the size of the hamartoma have any influence on the type of symptoms?
- Did all patients respond to GnRH treatment?
